# Dynamic structure of active nematic shells

Rui Zhang[1], Ye Zhou[1], Mohammad Rahimi[1] & Juan J. de Pablo[1]

When a thin film of active, nematic microtubules and kinesin motor clusters is confined on the surface of a vesicle, four $+1/2$ topological defects oscillate in a periodic manner between tetrahedral and planar arrangements. Here a theoretical description of nematics, coupled to the relevant hydrodynamic equations, is presented here to explain the dynamics of active nematic shells. In extensile microtubule systems, the defects repel each other due to elasticity, and their collective motion leads to closed trajectories along the edges of a cube. That motion is accompanied by oscillations of their velocities, and the emergence and annihilation of vortices. When the activity increases, the system enters a chaotic regime. In contrast, for contractile systems, which are representative of some bacterial suspensions, a hitherto unknown static structure is predicted, where pairs of defects attract each other and flows arise spontaneously.

[1] Institute for Molecular Engineering, University of Chicago, Chicago, Illinois 60637, USA. Correspondence and requests for materials should be addressed to J.J.d.P. (email: depablo@uchicago.edu).

A ctive systems consume and transform energy into local mechanical work at microscopic length scales[1,2]. Such systems arise in living cells, as is the case of the cytoskeleton[3], or can be realized *in vitro*, for example, in mixtures of biofilaments and their associated motor proteins[4–7]. At longer length scales, microswimmers[8,9] and even flocks of birds and schools of fish have been examined within the context of active systems[10,11]. Active materials can also be prepared from inorganic components, by relying on active colloids[12,13] or vibrating granular systems[14,15]. Of particular relevance to this work are recent experiments that have sought to elucidate the interplay between activity and geometric confinement. Examples include studies of cytoplasmic streaming[3], hydrodynamic instabilities[16,17], pattern formation[4,18] and counter rotating boundary layers[19]. Insightful qualitative arguments have been advanced to unravel the physics that underlie the above phenomena[20,21].

Many active systems consist of elongated molecules or assemblies; the dynamics of such systems are much more complex and potentially useful. To interpret their behaviour, one can rely on theoretical descriptions of nematic liquid crystals[22], which have been particularly helpful in elucidating a number of phenomena associated with active suspensions[23,24]. Despite the above successes, however, the ability to control ordered dynamics in active nematics has been limited and remains in its infancy[25,26].

Confinement has a profound effect on the collective dynamics of active systems. On the one hand, confinement dictates the structure of a nematic phase via topology and anchoring. For the particular case of a static nematic 'shell' confined by two concentric spherical surfaces (with degenerate planar anchoring), the ground-state configuration has four $+1/2$ disclination lines arranged into a tetrahedral configuration[27–29]. On the other hand, confinement can shape and stabilize spontaneous flows[19]. When taken together, these two factors lead to non-trivial phenomena. Simulations under appropriate boundary conditions, for example, suggest that active nematics in a capillary can develop bidirectional flows or helical vortices[30]. And recent calculations show that an active nematic drop on a surface can exhibit self-propulsion along well-defined directions[21]. In experiments, droplets of bacterial suspensions squeezed between two plates exhibit spontaneous circulation only for certain aspect ratios[19]. And a microtubule emulsion droplet under similar confinement may become motile[6].

When microtubules and kinesin are encapsulated within a shell, a new type of dynamic ordering emerges in which four $+1/2$ topological defects move in a well-defined pattern[25]. That ordering is particularly relevant for fundamental studies of active systems in that it provides a perfectly well-bounded, periodic system in which to interpret emerging views of active materials. The original experiments of Keber *et al.* were analysed within the framework of four coupled points (the defects) constrained to move on the surface of a sphere. That approach did not consider the molecular structure of the material explicitly, and in this work we rely on a continuum model of active nematics to describe and understand such a system. More specifically, we couple a Landau–de Gennes representation of a nematic liquid crystal to a hydrodynamic framework that accounts for activity to probe the effect of spherical-shell confinement on active nematics. The resulting patterns predicted by the model compare favourably with those observed experimentally[25], serving to validate the underlying theoretical treatment. The model is then used to elucidate the spatiotemporal details of the velocity and director fields, as well as the system's free energy. Recent experimental observations are interpreted in terms of distinct contributions to the free energy arising from enthalpy, 'bend' and 'splay'

deformation modes. For extensile systems at relatively low activity, defects move in closed trajectories. As activity increases, the system becomes unstable and enters a previously unreported chaotic regime that is characterized by open trajectories. In contrast, contractile systems, which are representative of some bacteria and for which experimental data are not yet available, yield a static defect structure at low and intermediate activities. The defects are attracted to each other in pairs, and the corresponding flow patterns produce intriguing stagnation points that could potentially be used for applications.

## Results

**Model system**. The active nematic shells considered here are confined between two concentric spherical surfaces that exhibit strong degenerate planar anchoring. A no-slip boundary condition is enforced on both surfaces. Note that the nematic coherence length $\xi_N$, which is usually comparable to the constituent's size, ranges from 1 (in microtubules)[6] to 10 μm (in bacteria)[9,31]. Here we choose $\xi_N = 2$ μm to describe the behaviour of a typical microtubule system. The shell is centred at the origin $O$, and has inner radius $R_{in} = 14$ and outer radius $R_{out} = 18$ in lattice units, which can be mapped onto a vesicle of radius 32 μm, comparable to that used in experiments. The timescale of the model is determined by choosing the characteristic length scale and the viscosity. By setting the rotational viscosity of the nematic material to $\gamma_1 = 0.1$ Pa s, the unit of time becomes $\tau = 4$ ms. The choice of elastic constant $K = 10$ pN is consistent with that adopted in numerous past studies of liquid crystalline systems[22,31]. The equilibrium scalar order parameter of the bulk, static nematic material is set to $q \simeq 0.62$, and all our calculations are restricted to the flow aligning regime by choosing a material constant $\xi = 0.8$, to reflect the fact that the aspect ratio of the typical biopolymer filaments in experiments is deep in the prolate regime. For instance, the *in vitro* microtubules used in refs 6,25 have a length-to-width ratio of $\simeq 60$. An initial static system is prepared via a Ginsburg–Landau relaxation[32]. A baseball-like director field is formed as the four defects, with topological charge $+1/2$, repel each other and adopt a tetrahedral arrangement[27–29]. To quantify the relative positions of the defects, an angular distance $\alpha_{ij}$ is introduced, given by the angle between radii $Oi$ and $Oj$, where $i$ and $j$ refer to the $i$th and $j$th defects, respectively. The average angular distance $\langle \alpha \rangle = \bar{\alpha}_{ij}$ represents the mean of the angles of the six defect pairs. At equilibrium, for a tetrahedral arrangement $\alpha_{ij} \equiv \alpha_0 \equiv 109.47°$, and therefore $\langle \alpha \rangle = 109.47°$.

The activity of the material is controlled by parameter $\zeta$. Hybrid lattice Boltzmann simulations are evolved for a duration of $t = 4 \times 10^6 \tau$, which corresponds to $\sim 1.6 \times 10^4$ s. We first focus on an extensile system with $\zeta > 0$. In Fig. 1, our simulation results are contrasted with the experimental images reported in ref. 25. Four different configurations are compared. Two snapshots are close to the ground state at which the four defects adopt a tetrahedral configuration (Fig. 1a,b,e,f). The other two snapshots are close to the excited state (Fig. 1c,d,g,h). As can be appreciated in the figure, our simulated images capture quantitatively the defect structure observed in experiments.

**Low-activity behaviour**. The system remains passive until the activity reaches a value of $\zeta \geq 0.0007$, at which point a spontaneous flow is generated. Below the onset activity, the defect configuration is deformed but remains static, as the elasticity balances the activity. Figure 2 shows a time sequence of representative images, separated by $\sim 80$ s, which reveal the position of the defects, along with the corresponding streamlines, for $\zeta = 0.001$. The highest velocities, which reach values as high as

$0.15\,\mu m\,s^{-1}$, are always associated with the defects, implying that the spatial gradient of the nematic order parameter induces the flows. The mean flow direction at the $+1/2$ defect is along its symmetry axis, which is consistent with experimental observations[23]. It is convenient to define this direction as the orientation of the defect. As can be appreciated in the figure, the

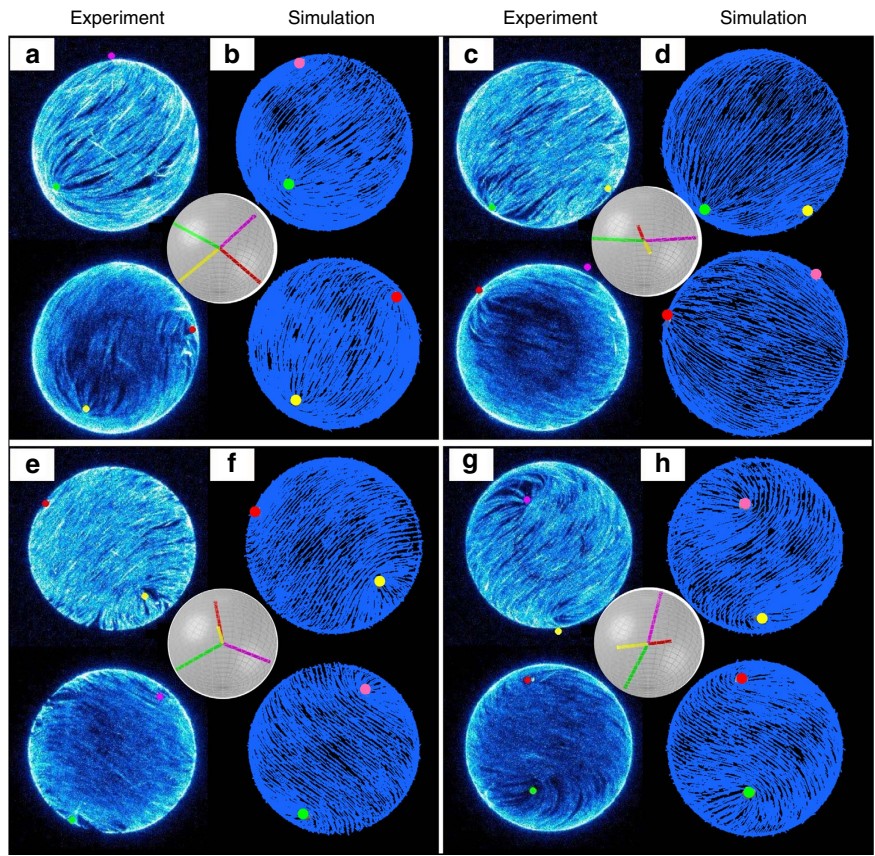

**Figure 1 | Representative configurations of active nematic shells at four different times.** They are labelled by **a,c,e,g** (experimental images from ref. 25, reprinted with permission from AAAS) and **b,d,f,h** (simulated structures). The two images within each labelled panel are the projections of opposite hemispheres. In all images, the coloured dots indicate the defect positions. In simulations, the blue lines correspond to the director field. The inset panels illustrate the defect configuration.

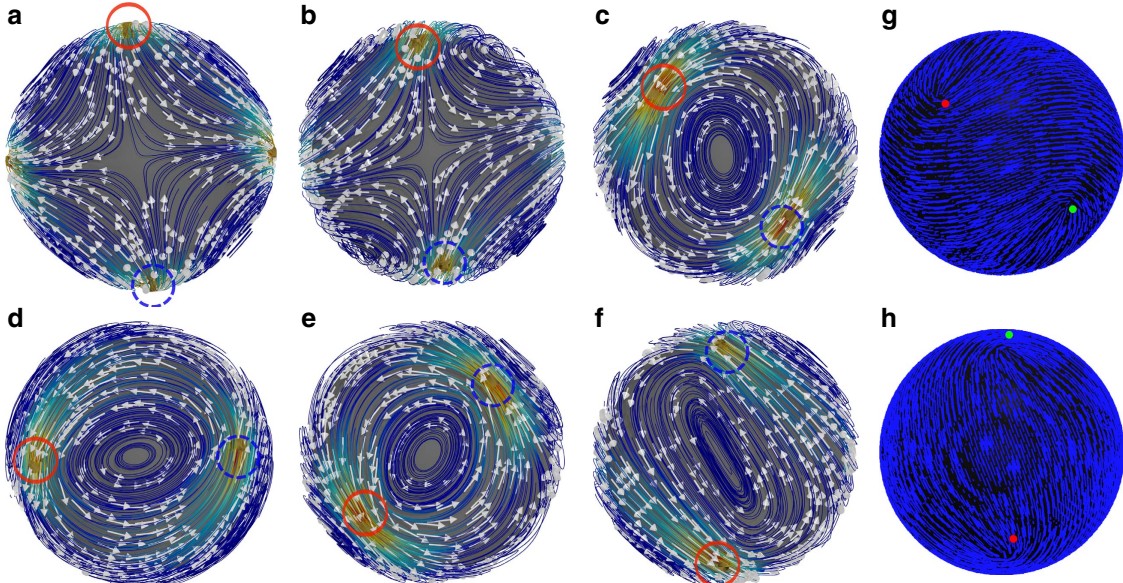

**Figure 2 | Vortex formation during evolution.** Time sequence (**a–f**) illustrate the velocity field of an active nematic shell for $\zeta = 0.001$. Defects are shown by yellow cylinders, velocity fields are shown by arrows and streamlines are shown by the white curved lines. The configurations shown here correspond to $t/\tau = 1.956 \times 10^6$ (**a**), $2 \times 10^6$ (**b**), $2.192 \times 10^6$ (**c**), $2.246 \times 10^6$ (**d**), $2.43 \times 10^6$ (**e**) and $2.482 \times 10^6$ (**f**). The positions of two of the defects are encircled. (**g,h**) The corresponding optical images and the director fields of **c,f**, respectively.

system develops vortices that are separated by defects. Occasionally (in Fig. 2a,c,e), four equally sized parallel eddies appear, with two extensional flows located at the two poles. Neighbouring vortices counter-rotate and, as shown later, this feature corresponds to an excited state.

Defect dynamics are controlled by two competing effects: one is the repulsive 'force' between the four, positively charged $+1/2$ defects, which has its origins in the elasticity of the material and drives the system towards its ground state. The other is the activity. The singularity of the director field at the defect serves to excite the system above its ground state. When one takes a derivative of equation (11), at rest, the last term of that equation does not vanish, leading to the spontaneous emergence of flow. It is found that for the range of activity between $0.0007 \leq \zeta \leq 0.005$, the four defects move in closed trajectories along the edges of a deformed cube. One can observe that each pair of defects is symmetric about the symmetry axis of the cube, and therefore $\alpha_{12} = \alpha_{34}$, $\alpha_{13} = \alpha_{24}$ and $\alpha_{14} = \alpha_{23}$ (Supplementary Note 1), implying that only three independent angular distances are necessary to describe the configuration of the defects. For conciseness, we use $\alpha_1$, $\alpha_2$ and $\alpha_3$ to denote these and thus $\langle \alpha \rangle = (\alpha_1 + \alpha_2 + \alpha_3)/3$.

The results in Fig. 3, for $\zeta = 0.001$, serve to explain the system's entire motion mechanism. As the four defects move collectively, their configurations oscillate between a tetrahedral mode and a planar mode. When they move to the four corners of the deformed cube, they form a tetrahedron. The defect–defect distance is at its maximum, and $\alpha_1 = \alpha_2 = \alpha_3 = \langle \alpha \rangle \equiv \alpha_0 = 109.47°$. As shown in Fig. 3d, the system is in its free-energy ground state. When the defects move to the midpoints of the four parallel sides of the cube, they form a square within a single plane. At that point $\alpha_1 = 180°$ and $\alpha_2 = \alpha_3 = 90°$, and the

resultant average angular distance becomes $\langle \alpha \rangle \equiv 120°$. In this mode, the defect–defect distance is on average smaller than that in the tetrahedral configuration. The system's free energy reaches a maximum, as shown in Fig. 3d, indicating that this arrangement corresponds to an excited state. For $\zeta = 0.004$, the system oscillates between these two modes with period $T = 3.14 \times 10^4 \tau = 125$ s. This estimate is consistent with the experimental value reported in ref. 25, and it is established by the system size and the average flow velocity. When the system is in the tetrahedral mode, the free energy is close to that of the static system. There are, however, two differences between the ground state of the extensile active system and the static state of a nematic. One is that the ground state of the extensile active system cannot be a short-arc state (if one draws a geodesic line connecting the two defects following the director field, the curve is a short arc[33]). As the defects try to move along their orientations (symmetry axis) they necessarily form a long-arc state (if one draws a geodesic line connecting the two defects following the director field, the curve is a long arc). As shown later, for a contractile system the configuration always consists of a short-arc state. The other difference is that the orientations of the defect pairs are not directly 'against' each other. As the system passes the tetrahedral state, two approaching defects deflect of each other at an angle due to elastic repulsions. As shown in Fig. 2g, this deflection requires that the orientations of the defects do not point towards each other on the sphere.

To elucidate the precise origin of the oscillatory nature of the free energy, in Fig. 3e we plot the different contributions to the free energy of the system. Note that for our quasi-two-dimensional geometry, the twist is 0 and the saddle-splay is time invariant. We therefore only show splay, bend and defect (phase) energies. The defect (phase) energy is the Landau–de Gennes

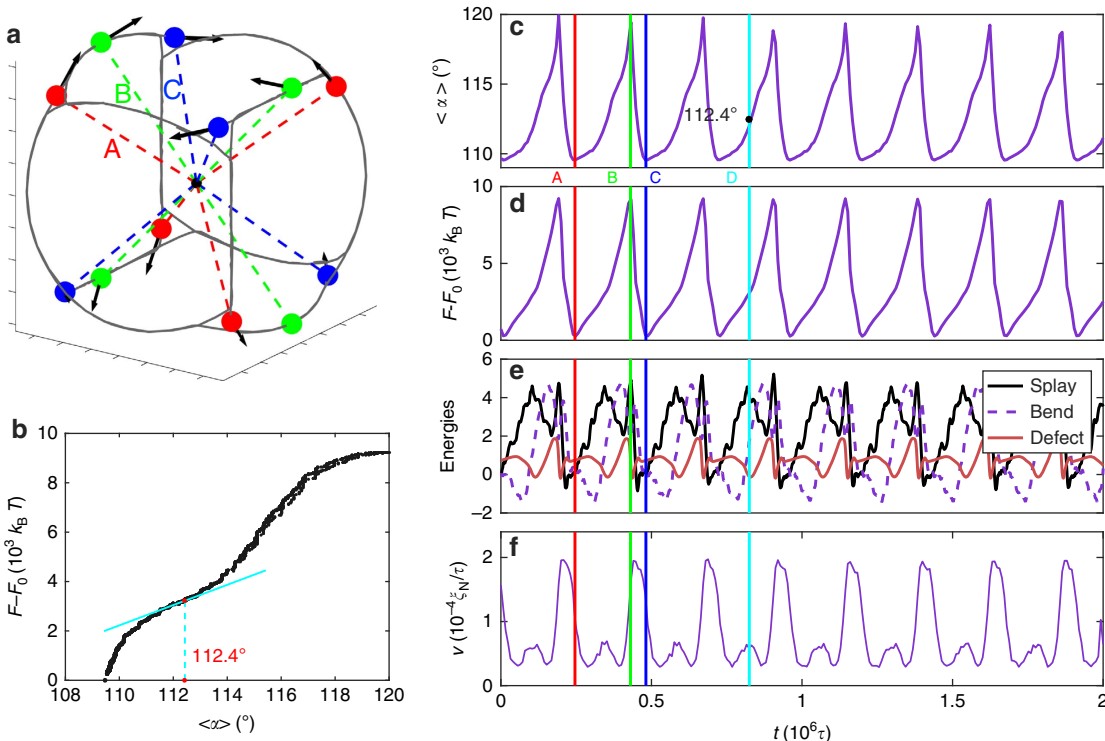

**Figure 3 | Defect configurations and trajectories for low activity. (a)** Defect positions for (filled circles) $\zeta = 0.001$ at three consecutive times, marked by A, B and C. The defects' orientations are illustrated by arrows, and the dashed lines connect the defects and the origin. **(b)** Correspondence between the free energy $F$ and $\langle \alpha \rangle$. **(c)** Time evolution of the mean angular distance; **(d)** free energy of the system; **(e)** splay, bend and defect energies, referenced by initial value; **(f)** mean velocity of the four defects. The static system's free energy $F_0$ is used as a reference. The three vertical lines mark the times of the three snapshots. The fourth vertical line $d$ marks a secondary maximum in the velocity plot of **f**.

short-range free-energy term $f_p$ defined in equation (5). When the system is in its ground state, the splay and bend energies are relatively low. In contrast, when the system is in its excited state, the splay energy reaches its peak value, and the bend energy is relatively high. There is a phase shift between the splay and bend contributions. We also note that the defect energy exhibits a sharp decline right after the system passes the energy barrier, as the release of defect core energy drives defect motion towards the ground state. Interestingly, when the velocity reaches a secondary peak (shown in Fig. 3f), the phase energy goes to a minimum but the splay contributions reach a secondary peak. It is also of interest to examine the evolution of the spatial distribution of different contributions to the free energy. The relevant distributions are shown and discussed in Supplementary Note 2 (Supplementary Fig. 1), where one can appreciate the formation of a low-splay band between pairs of defects.

It is worth pointing out here that the effect of biopolymer flexibility can be included implicitly in our model through a reduced bend elastic constant. When the activity is sufficiently strong, however, the filaments may buckle and lose their rod-like shape. In such cases, a more elaborate treatment of filament flexibility is required. We leave this issue and the possible effects of disparate elastic constants for subsequent studies.

The temporal evolution curves of $\langle\alpha\rangle$ and the system's free energy $F$ are ratchet like. The defect velocity is plotted in Fig. 3f. When the system moves from the ground (tetrahedral) state towards the excited, planar state (between time points A and B in Fig. 3), the dynamics becomes slow as the system gains potential energy. However, when the system is moving from an energetically unfavourable state towards a ground state (between time points B and C in Fig. 3), the defect velocity reaches a maximum, and the temporal curves of both $\langle\alpha\rangle$ and $E_{el}$ show sharp declines. If $t_A$, $t_B$ and $t_C$ are used to denote the times corresponding to the three time points shown in Fig. 3, the percent of time spent by the system climbing the free-energy barrier $\beta = (t_B - t_A)/(t_C - t_A)$ can be used to characterize the asymmetry of the curves; for $\zeta = 0.001$, we have $\beta = 0.78$. The asymmetry in our dynamics could potentially be useful for engineering purposes; for instance, during self propulsion, as in the case of an active nematic vesicle, the asymmetric oscillation of the defects could be used to induce directional motion. A secondary peak in the velocity plot can also be appreciated, labelled as time point D in Fig. 3. If one plots the system's free energy $F$ and its associated $\langle\alpha\rangle$ at different times on a single figure, those data collapse onto a single curve that describes the free-energy landscape in terms of $\langle\alpha\rangle$ (Fig. 3b). The slope of such a curve shows a secondary minimum at $\alpha = 112.4°$, which is the exact angle when the system reaches a secondary defect velocity maximum.

Our simulation results at low activity agree with the phenomenological model proposed in ref. 25 in the following respects: (1) both models predict a ratchet-like shape when represented in a $\alpha$-plot. As explained above, that shape is manifestation of the interplay between activity and elasticity. (2) Both models exhibit a threshold/onset activity, below which the system cannot overcome the elasticity to enter the oscillatory dynamic state. However, the defect trajectories predicted by the two models are different. In the treatment presented in ref. 25, the defects form pairs, and the paired defects revolve around the pair's centre of mass[25]. In our simulations and in the

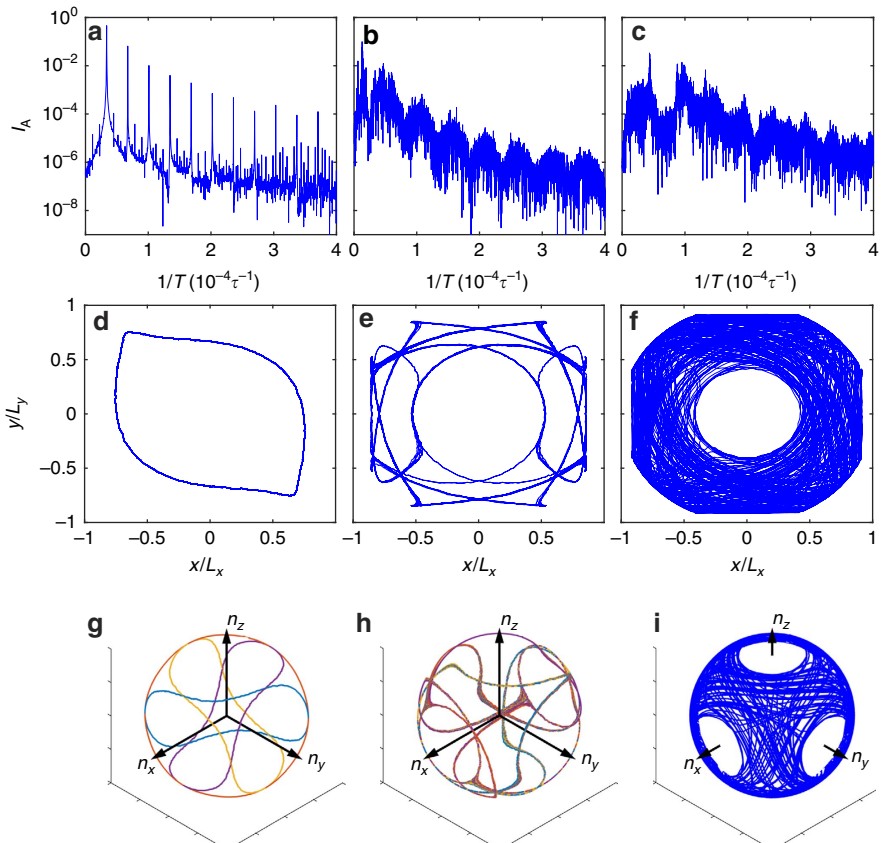

**Figure 4 | Spectral analysis of defect configurations.** System evolution from periodic (**a,d,g**, with $\zeta = 0.0042$) to quasiperiodic (**b,e,h**, with $\zeta = 0.0052$) and chaotic (**c,f,i**, with $\zeta = 0.01$). (**a–c**) Power spectrum of the time series of $\langle\alpha\rangle$. (**d–f**) Projection of one defect trajectory onto the $xy$ plane. (**g–i**) The four defect trajectories in three dimensions. Trajectories in **g,h** are made transparent and that in **i** are not to assist eyes.

experiments, defect trajectories exhibit a more complicated behaviour: the defects do not form pairs, and their trajectories are not simple circles. Instead, the defects can occasionally turn by $\sim 90°$ during motion.

The vesicle in the experiments can be made more flexible by changing the hypertonic stress. Experiments show that the $+1/2$ defects on a flexible vesicle can grow protrusions[25]. We think this phenomenon arises from the interplay between the dynamics of the $+1/2$ defect, the curvature of the vesicle and the excess surface area provided by the hypertonic stress. As the comet-like $+1/2$ defect moves, the microtubules at the tail of the defect move along the tangential plane of the vesicle, but the curvature forces the motion to bend, and follow the sphere's surface. This yields an outward stress that is able to protrude the vesicle.

**Intermediate- and high-activity behaviour.** We now consider how the system evolves under high activity. In Supplementary Figs 2 and 3, we show the defect trajectories for intermediate activities. As explained below, a power spectrum analysis reveals a transition from periodic to quasi-periodic and to chaotic dynamics as $\zeta$ increases[34]. When $\zeta$ is gradually raised to 0.005, the cube deforms and the asymmetry in the $\langle \alpha \rangle$ plot fades away. The above trend is discussed in Supplementary Note 3. When $\zeta > 0.005$, the $\langle \alpha \rangle$ plot no longer consists of a single periodic oscillation (Supplementary Fig. 3 for details). When $\zeta \geq 0.006$, the defect trajectories become open (Supplementary Fig. 4). Figure 4a–c shows the power spectrum of the time series of $\langle \alpha \rangle$ for low ($\zeta = 0.0042$), medium ($\zeta = 0.0052$) and high activity ($\zeta = 0.01$). At $\zeta = 0.0042$, the defect trajectories in the two-dimensional and three-dimensional plots (Fig. 4d,g) are closed, and the power spectrum shows sharp peaks corresponding to their oscillation period and its harmonics. When $\zeta = 0.0052$, the defect trajectories are still closed but exhibit a more intricate geometry (Fig. 4e,h). The corresponding power spectrum exhibits equally spaced frequencies, but with significant noise in between. For $\zeta = 0.01$, the defect trajectories are chaotic outside a depletion region (Fig. 4f,i, see Supplementary Note 4 for discussion). The power spectrum shows no evidence of periodicity, and in Supplementary Fig. 4 we further show that the chaotic system is ergodic.

Importantly, the oscillation period $T$ is highly sensitive to activity $\zeta$. As shown in Fig. 5c, $1/T$ is linear in $\zeta$ for $0.0007 \leq \zeta \leq 0.005$, the closed-trajectory regime, where $T$ varies from $2.16 \times 10^6 \tau$ to $2.5 \times 10^4 \tau$. This dependence could in fact be used to either measure the macroscopic quantity $\zeta$ or to quantify the concentration of ATP in experiments. When the system transitions to the less-ordered state, in the range

$0.006 \leq \zeta \leq 0.01$, one can still measure a 'period' $T$ by defining it as the average time spent between two consecutive planar modes (illustrated in Supplementary Fig. 2d,e). For that measure, $1/T$ decreases slightly but is still linear with respect to $\zeta$. The inset of Fig. 5c shows that the underlying characteristic flow velocity is approximately proportional to $\zeta$, which is consistent with literature reports[6,24]. The transition of the period plot from one linear regime to another is due to the transition of the system from periodic to chaotic dynamics.

**Contractile system.** We conclude with a discussion of a contractile system, for which $\zeta < 0$. When $\zeta \geq -0.05$, the system comes to rest (in terms of its defect structure) after a few oscillations. The steady state corresponds to four defects forming two pairs; within each pair, the defects are attracted to each other. As shown in Fig. 6b, the two pairs appear at the two poles (one pair is not shown for clarity). They are oriented back to back, forming a short arc. Although the director field and defect configuration are stationary, a velocity field still exists (Fig. 6a). It is primarily localized near the defect pairs. The mean flow direction at the defect is opposite to its orientation, in contrast to extensile systems, where the mean flow direction is along the defect orientation. Extensional flows emerge between the defect pair, a feature that could be useful for design of microfluidic applications. In this case, the activity drives the system to a state from which it cannot escape. The activity tries to attract two $+1/2$ defects, but this is prevented by elasticity. As shown in Fig. 6d, as activity increases in a contractile state, the angular distance between attracted defects at steady state becomes smaller, and the system's free energy increases. For even more negative values of $\zeta$, the four-defect structure becomes unstable. Occasionally, it fuses into two $+1$ defects located at the poles, forming a bipolar structure. But this state is fragile, the system evolves into a multi-defect configuration, and momentum is no longer conserved. We are not aware of contractile biopolymer systems, despite the fact that some materials are contractile-like[7]. However, non-nematic micro-swimmer systems, such as algae, can be contractile. We therefore propose that future experiments on bacterial swimmers in nematic shells could be used to assess the merits of our predictions and, if correct, could be used as the basis for creation of self-propelled microfluidic devices capable of producing both controlled shear and elongational forces.

We conclude by noting that, to address any possible size effects, we have performed simulations for system with $R_{in} = 34$ and $R_{out} = 38$. The results for large systems are similar to those for their smaller counterparts, but, as expected, the transition value of the activity $\zeta$ is different.

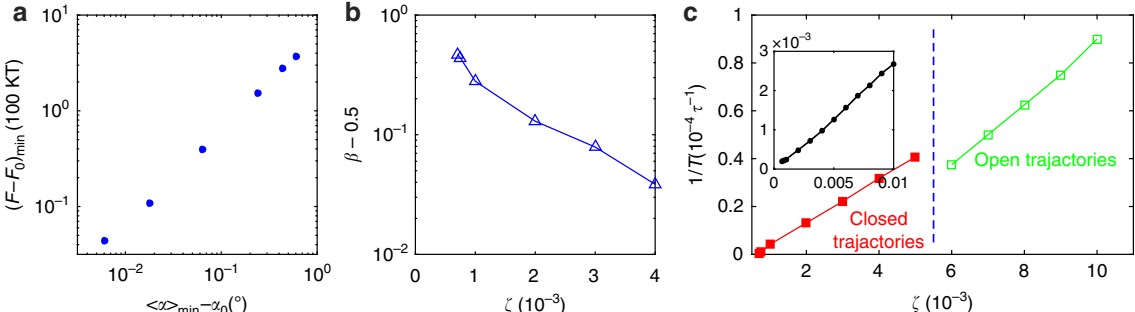

**Figure 5 | Activity dependence plots.** (**a**) Log–log scatter plot of the minimum of $\langle \alpha \rangle - \alpha_0$ and the minimum of $F - F_0$ for $0.0007 \leq \zeta \leq 0.004$; (**b**) Shape asymmetry $\beta = (t_B - t_A)/(t_C - t_A)$ of the angular distance curve on activity $\zeta$; $t_A$, $t_B$ and $t_C$ are the times of three consecutive tetrahedral, planar and tetrahedral modes, respectively, as illustrated in Fig. 3; (**c**) Oscillation period $T$ as a function of $\zeta$. The inset shows the maximum flow velocity ($\xi_N/\tau$) at the defect versus $\zeta$.

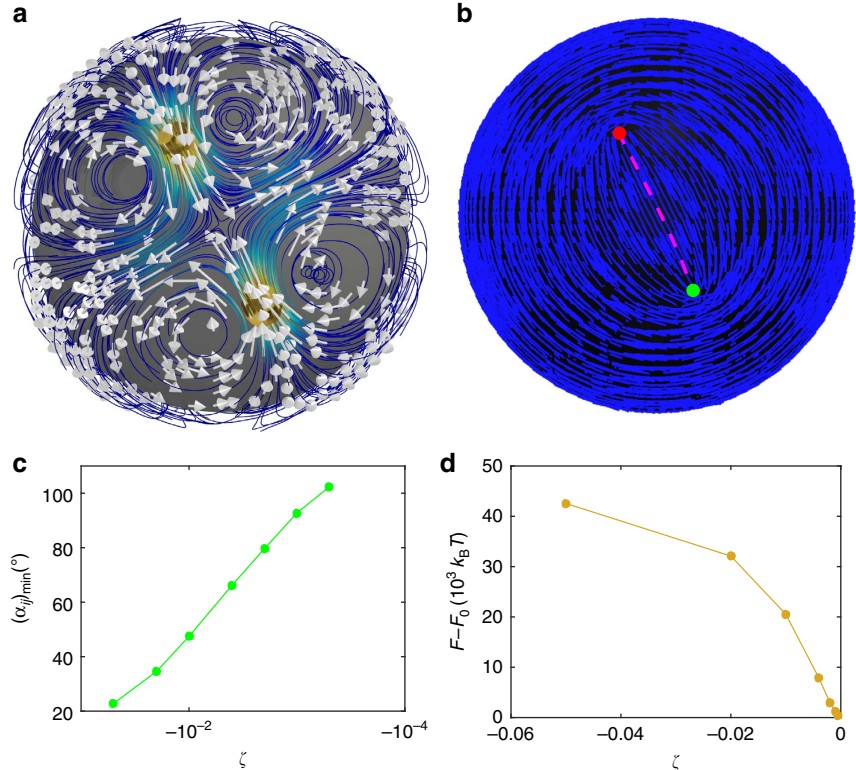

**Figure 6 | Contractile system structure and dynamics.** Streamlines (**a**), defects (cylinders) and director field (**b**) of contractile system for $\zeta = -0.007$ at steady state. In **b**, the curve connecting the two defects along the director field is a short arc. The angular distance between these defects is a minimum. Activity dependence of the minimum angular distance $(\alpha_{ij})_{min}$ (**c**) and free energy $F - F_0$ at steady state (**d**).

## Discussion

Recent experiments with microtubule filaments and kinesin motor clusters encapsulated in shells have revealed the existence of intriguing structures and closed-loop trajectories[25]. In this work, a theory for active nematics, coupled to the relevant momentum conservation equations, has been used to explain their temporal evolution. In agreement with experiment, the theory predicts the formation of a four-defect structure that oscillates periodically between a tetrahedral configuration (ground state) and a planar configuration (excited state), thereby lending support to the proposed model. It is then shown that the underlying physics leading to the observed periodic oscillations of structure is the competition between elasticity and activity. More specifically, a complex interplay between splay, bend and defect energies is identified in which, in the ground state, the splay and bend energies have a near-minimal value but the defect adopts an intermediate size. In contrast, in the excited state, splay deformations, as well as the size of the defect cores, reach a maximum. The corresponding velocity of the defects is also near its maximum value at that point.

For low to intermediate activity, the four-defect structure is stable. In extensile systems, a moving defect induces a flow, and the mean flow direction follows the defect's orientation. For low activity, defect trajectories form a closed loop that can be mapped onto the edges of a deformed cube. This fact can be used to rationalize the oscillating behaviour of the defect configuration. As the activity increases, the closed trajectories deform and the oscillation frequency increases linearly. This observation suggests that one may in fact use such a dependence to measure the macroscopic activity. By increasing the activity even further, the defect trajectories open up and they enter an ergodic state outside the depletion region, which is defined by the symmetry axes of the cube. In contrast, for contractile systems, which are representative of bacteria or algae, the mean flow direction at a defect is opposite to its orientation. Such flows correspond to an intriguing static state of the director field, where defects are attracted to each other in pairs, forming a short-arc state.

## Methods

**Governing equations.** A hybrid lattice-Boltzmann method is used to simultaneously solve a modified Beris–Edwards equation and a momentum equation, which account for the activity of the nematic material. The nematic phase is described by a tensorial order parameter $\mathbf{Q} = \langle \mathbf{nn} - \frac{1}{3}\mathbf{I} \rangle$, where unit vector $\mathbf{n}$ describes the director orientation, $\mathbf{I}$ the identity tensor and $\langle \rangle$ denotes an ensemble average. By introducing a velocity gradient $W_{ij} = \partial_j u_i$, $\mathbf{A} = (\mathbf{W} + \mathbf{W}^{\mathbf{T}})/2$, $\Omega = (\mathbf{W} - \mathbf{W}^{\mathbf{T}})/2$, and a generalized advection term

$$\mathbf{S}(\mathbf{W}, \mathbf{Q}) = (\xi\mathbf{A} + \Omega)(\mathbf{Q} + \mathbf{I}/3) + (\mathbf{Q} + \mathbf{I}/3)(\xi\mathbf{A} - \Omega) - 2\xi(\mathbf{Q} + \mathbf{I}/3)\mathrm{Tr}(\mathbf{QW}). \tag{1}$$

one can write a modified Beris–Edwards equations[35,36] according to

$$(\partial_t + \mathbf{u} \cdot \nabla)\mathbf{Q} - \mathbf{S}(\mathbf{W}, \mathbf{Q}) = \Gamma\mathbf{H} + \lambda\mathbf{Q}. \tag{2}$$

The constant $\xi$ is related to the material's aspect ratio, and $\Gamma$ is related to the rotational viscosity $\gamma_1$ of the system by $\Gamma = 2q_0^2/\gamma_1$ (ref. 37), where $q_0$ is the scalar order parameter of the nematic phase. In equation (2), $\lambda$ represents the first activity parameter, which is equivalent to varying the static nematic order parameter[36]. The molecular field $\mathbf{H}$, which drives the system towards thermodynamic equilibrium, is given by

$$\mathbf{H} = -\left[\frac{\delta F}{\delta \mathbf{Q}}\right]^{\mathrm{st}}, \tag{3}$$

where $[\ldots]^{\mathrm{st}}$ is a symmetric and traceless operator, and $F$ is the total free energy of the system, defined by

$$F = \int_{\mathrm{bulk}} \mathrm{d}V f_{\mathrm{bulk}} + \int_{\mathrm{surface}} \mathrm{d}S f_{\mathrm{surf}}. \tag{4}$$

The terms $f_{\mathrm{bulk}}$ and $f_{\mathrm{surf}}$ represent the bulk and surface contributions to the free energy, respectively. Here $f_{\mathrm{bulk}} = f_{\mathrm{p}} + f_{\mathrm{e}}$, where $f_{\mathrm{p}}$ is the short-range or 'phase'

energy and $f_e$ is the long-range elastic energy. The phase energy $f_P$ is given by a Landau–de Gennes expression of the form refs 38,39

$$f_P = \frac{A_0}{2}\left(1 - \frac{U}{3}\right)\mathrm{Tr}(\mathbf{Q}^2) - \frac{A_0 U}{3}\mathrm{Tr}(\mathbf{Q}^3) + \frac{A_0 U}{4}\left(\mathrm{Tr}(\mathbf{Q}^2)\right)^2. \qquad (5)$$

Parameters $U$ and $\lambda$ control the magnitude of $q_0$ via ref. 36

$$q_0 = \frac{1}{4} + \frac{3}{4}\sqrt{1 - \frac{8}{3U}\left(1 - \frac{\lambda}{\Gamma A_0}\right)}. \qquad (6)$$

The nematic coherence length, given by $\xi_N = \sqrt{K/A_0}$, determines the size of a defect core and serves as the fundamental length scale for our description of nematic materials. In our nomenclature, $f_P$ provides a measure of the total core energy of the defects that arise in the system.

The elastic energy $f_e$ is written as

$$f_e = \frac{1}{2}L_1 Q_{ij,k}Q_{ij,k} + \frac{1}{2}L_2 Q_{jk,k}Q_{jl,l} + \frac{1}{2}L_3 Q_{ij}Q_{kl,i}Q_{kl,j} + \frac{1}{2}L_4 Q_{ik,l}Q_{jl,k}. \qquad (7)$$

The precise connection between this free-energy expression and the common Frank elasticity theory is discussed in the following subsection.

Degenerate planar anchoring is implemented through a Fournier–Galatola expression[40] that penalizes out-of-plane distortions of the $\mathbf{Q}$ tensor. The associated free-energy expression is given by

$$f_{surf} = W\left(\tilde{\mathbf{Q}} - \tilde{\mathbf{Q}}^\perp\right)^2, \qquad (8)$$

where $\tilde{\mathbf{Q}} = \mathbf{Q} + (q_0/3)\mathbf{I}$ and $\tilde{\mathbf{Q}}^\perp = \mathbf{P}\tilde{\mathbf{Q}}\mathbf{P}$. Here $\mathbf{P}$ is the projection operator associated with the surface normal $v$ as $\mathbf{P} = \mathbf{I} - vv$. The evolution of the surface $\mathbf{Q}$-field is governed by ref. 41

$$\frac{\partial \mathbf{Q}}{\partial t} = -\Gamma_s\left(-Lv\cdot\nabla\mathbf{Q} + \left[\frac{\partial f_{surf}}{\partial \mathbf{Q}}\right]^{st}\right), \qquad (9)$$

where $\Gamma_s = \Gamma/\xi_N$. The above equation is equivalent to the mixed boundary condition given in ref. 42 for steady flows.

Using an Einstein summation rule, the momentum equation for the active nematics can be written as

$$\rho\left(\partial_t + u_\beta\partial_\beta\right)u_\alpha = \partial_\beta\Pi_{\alpha\beta} + \eta\partial_\beta\left[\partial_\alpha u_\beta + \partial_\beta u_\alpha + (1 - 3\partial_\rho P_0)\partial_\gamma u_\gamma\delta_{\alpha\beta}\right]. \qquad (10)$$

The stress $\Pi$ is defined as

$$\begin{aligned}\Pi_{\alpha\beta} = \ &{-P_0\delta_{\alpha\beta}} - \xi H_{\alpha\gamma}\left(Q_{\gamma\beta} + \tfrac{1}{3}\delta_{\gamma\beta}\right) - \xi\left(Q_{\alpha\gamma} + \tfrac{1}{3}\delta_{\alpha\beta}\right)H_{\gamma\beta}\\ &+ 2\xi\left(Q_{\alpha\beta} + \tfrac{1}{3}\delta_{\alpha\beta}\right)Q_{\gamma\epsilon}H_{\gamma\epsilon} - \partial_\beta Q_{\gamma\epsilon}\frac{\delta\mathcal{F}}{\delta\partial_\alpha Q_{\gamma\epsilon}}\\ &+ Q_{\alpha\gamma}H_{\gamma\beta} - H_{\alpha\gamma}Q_{\gamma\beta} - \zeta Q_{\alpha\beta},\end{aligned} \qquad (11)$$

where $\eta$ is the isotropic viscosity, and the hydrostatic pressure $P_0$ is given by ref. 43

$$P_0 = \rho T - f_{bulk}. \qquad (12)$$

The temperature $T$ is related to the speed of sound $c_s$ by $T = c_s^2$. The second activity parameter, $\zeta$, accounts for the local stress that arises from spatial gradients of the nematic order parameter[1,36,44]. If $\zeta > 0$, the system is extensile. If $\zeta < 0$, it is contractile.

We solve the evolution equations, equations (2) and (9), using a finite-difference method. The momentum equation, equation (10), is solved simultaneously via a lattice Boltzmann method over a D3Q15 grid[45]. The implementation of stress follows the approach proposed by Guo et al.[46]. Our model and implementation were validated by comparing our simulation results to predictions using the Ericksen-Leslie-Parodi (ELP) theory[22,47–49] in the absence of activity. We refer the reader to ref. 41 for additional details on the numerical methods used here.

**Elastic constant mapping.** Given that the system's director field is described by a unit vector field $\mathbf{n}$, the Frank–Oseen expression for elastic energy density $f_e$ reads

$$\begin{aligned}f_e = \ &\frac{1}{2}K_{11}(\nabla\cdot\mathbf{n})^2 + \frac{1}{2}K_{22}(\mathbf{n}\cdot\nabla\times\mathbf{n})^2 + \frac{1}{2}K_{33}(\mathbf{n}\times(\nabla\times\mathbf{n}))^2\\ &- \frac{1}{2}K_{24}\nabla\cdot[\mathbf{n}(\nabla\cdot\mathbf{n}) + \mathbf{n}\times(\nabla\times\mathbf{n})],\end{aligned} \qquad (13)$$

where $K_{11}$, $K_{22}$, $K_{33}$ and $K_{24}$ refer to splay, twist, bend and saddle-splay moduli, respectively. If the system is uniaxial, the $L$'s in equation (7) can be determined through

$$\begin{aligned}L_1 &= \tfrac{1}{2q_0^2}\left[K_{22} + \tfrac{1}{3}(K_{33} - K_{11})\right],\\ L_2 &= \tfrac{1}{q_0^2}(K_{11} - K_{24}),\\ L_3 &= \tfrac{1}{2q_0^2}(K_{33} - K_{11}),\\ L_4 &= \tfrac{1}{q_0^2}(K_{24} - K_{22}).\end{aligned} \qquad (14)$$

By adopting a one-elastic-constant approximation, $K_{11} = K_{22} = K_{33} = K_{24} \equiv K$, one has $L_1 = L \equiv K/2q_0^2$ and $L_2 = L_2 = L_2 = 0$. Point wise, $\mathbf{n}$ is the eigenvector associated with the greatest eigenvalue of the $\mathbf{Q}$-tensor at each lattice point. The derivatives of $\mathbf{n}$ are obtained via a finite-difference method. To avoid singularities,

we calculate the elastic energies on bulk points with order parameter $q > 0.45$ ($q = q_0 \simeq 0.62$ for undistorted nematics and $q \simeq 0.2$ at defect cores).

**Data availability.** Data and analysis codes are available from the authors upon request.

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

## Acknowledgements

The development of the non-equilibrium lattice Boltzmann method presented here for active nematics was supported by NSF DMR-1410674. The analysis of active nematic materials was supported by the University of Chicago Materials Research Science and Engineering Center (NSF DMR-1420709). We are grateful for the support of the University of Chicago Research Computing Center for assistance with the calculations carried out in this work. We thank Prof Julia Yeomans, Prof M. Cristina Marchetti, Prof Miha Ravnik, Prof Daniel J. Needleman, Prof Margaret Gardel, Dr Stephen J. DeCamp, Dr Arnout Boelens, Dr Abelardo Ramirez-Hernandez and Zhihong You for helpful discussions.

## Author contributions

R.Z. and J.J.d.P designed the research; R.Z. and J.J.d.P. performed the research; R.Z., Y.Z., M.R. and J.J.d.P. analysed the data; R.Z. and J.J.d.P. wrote the paper; J.J.d.P. supervised the research. All authors discussed the progress of research and reviewed the manuscript.

## Additional information

**Competing financial interests:** The authors declare no competing financial interests.

