## [Peer Review File · Nature Communications]

Reviewers' comments:

Reviewer #1 (Remarks to the Author):

In the manuscript entitled "Dynamic structure of active nematic shells" describes a numerical study of active nematic crystals confined onto a spherical shell. The study presents a significant advance over the previous model that has described motile defect in active nematic shells as active particles that interact through effective repulsions. The authors describe a number of regimes, which exhibit intriguing dynamical behaviors. Only some of these have been observed experimentally. The subject is very timely, the manuscript is well written and I fully support its publication in Nature Communications.

It seems that the formalism developed by the authors could be extended to other ever more interesting cases. For example, it would be interesting to systematically increase the diameter of the spherical shell. Above certain critical diameter one should observed spontaneous creation of defect pairs. This regime has not been explored either from experimental or theoretical perspective. Also one could apply the formalism developed in this manuscript to other topologies such as a toroid in which the defect dynamics would couple to the background curvature. I am not suggesting that these studies need to be included in the present work, but they would certainly be interesting to pursue in the future.

The manuscript presentation could be improved by considering following minor suggestions.

1. In Figure 3c3 the authors plot the defect energy. This quantity should be defined and described in more detail.
2. It would be useful to discuss and compare the results of full numerical calculation to the much simpler model that treats motile defects as active repulsive particles. Is the simpler particle based model valid in any regime?
3. Is there a critical lower activity required for the onset of oscillations? If yes what happens below this lower activity.
4. On line 181 authors discuss short arc and long arc states. Perhaps these can be defined somewhere.
5. One page 10 authors state: "We propose that the planar mode has high symmetry, and therefore the penalty arising from bending is at a local minimum." Perhaps this can be explained in more detail.
6. Do authors examine the influence of elastic constants that differ in magnitude? If not this is another area worthy of future investigation.

Reviewer #2 (Remarks to the Author):

The authors use a continuum model of an active nematic to model active flow in a shell. This simulation approach has been used to describe active nematics in several contexts and gives good agreement with experiments. What is new here is handling the shell geometry within this simulation approach.

The work is motivated by experiments where microtubules and kinesin motors are confined on the surface of a vesicle. The combination of nematic order and the vesicle geometry imposes four topological defects on the surface of the vesicle and the activity drives these to move around the shell in orbits that are regular at low activities but become chaotic at high activities.

The current paper reproduces this behaviour well and gives many details of the competition between elasticity, flow and activity that control the defect dynamics. Defect trajectories are plotted as a function of activity and the balance between the various contributions to the free energy are explained. So this is a useful addition to the literature and a very careful piece of work. However, I think it would be much better published in a more specialised journal as the physics underlying the defect dynamics is explained in refs [6] and [24]. Moreover, there is an overlapping paper that the authors have missed: Motility of active fluid drops on surfaces, Diana Khoromskaia and Gareth P. Alexander Phys. Rev. E 92, 062311 (2015).

Minor points:

1. Does the thickness of the shell map onto the thickness of the vesicle, and does it affect the results?
2. Why is the material flow aligning, would flow tumbling dynamics affect the results?
3. The authors refer to their model as a 'molecular' model. I would rather call it a coarse-grained or continuum model as there is no molecular detail.
4. It would be interesting if the authors could comment on the likely effect of the flexibility of the microtubules which is not included in the model.
5. ... and on the flexibility of the vesicle - in the experiments protrusions sometimes form at the defect sites and it would be interesting (although difficult and not expected in the current M/S) to understand why.
6. p2 It is worth noting that nematic symmetry can arise from the flow field, not necessarily the shape of the active elements.
7. p2 I do not agree with the statement 'confinement can induce and stabilise nematic flows' - confinement can stabilise the flow, but not induce it.
8. ref [24] typo in Dogic

Reviewer #3 (Remarks to the Author):

In the present manuscript the authors simulate an active nematic shell. With the help of a hybrid Boltzmann method they are able to simulate different regimes of activity. In the low activity state they recover amazingly precisely the activity observed in experiments, at higher activity they observe a chaotic regime. They are able to distinguish the different contributions to the free energy of the system - it is the interplay of enthalpy, bend and splay modes which gives rise to the unique dynamics.

The manuscript is beautifully written, and the approach as well as the results are bathing the way for new exciting studies of this class of materials. I do recommend enthusiastically the acceptance of the manuscript as it is. I do not see how to improve the manuscript any further.

REVIEWERS' COMMENTS:

Reviewer #1 (Remarks to the Author):

The authors have addressed all my comments and suggestions. I fully support the publication of the manuscript in its current form.

Zvonimir Dogic

Reviewer #2 (Remarks to the Author):

I think the authors have justified the novelty of their manuscript in their reply pointing out things that I had missed. Therefore I am pleased to recommend publication.

Point-by-point response to the referees' comments

We thank the reviewers for their helpful comments. Please find our response as follows.

Reviewer #1 (Remarks to the Author):

In the manuscript entitled "Dynamic structure of active nematic shells" describes a numerical study of active nematic crystals confined onto a spherical shell. The study presents a significant advance over the previous model that has described motile defect in active nematic shells as active particles that interact through effective repulsions. The authors describe a number of regimes, which exhibit intriguing dynamical behaviors. Only some of these have been observed experimentally. The subject is very timely, the manuscript is well written and I fully support its publication in Nature Communications.

It seems that the formalism developed by the authors could be extended to other ever more interesting cases. For example, it would be interesting to systematically increase the diameter of the spherical shell. Above certain critical diameter one should observed spontaneous creation of defect pairs. This regime has not been explored either from experimental or theoretical perspective. Also one could apply the formalism developed in this manuscript to other topologies such as a toroid in which the defect dynamics would couple to the background curvature. I am not suggesting that these studies need to be included in the present work, but they would certainly be interesting to pursue in the future.

Response:

We appreciate the Referee's positive and constructive comments. One of the advantages of our method is that it can be easily extended to different geometries. As the curvature decreases (vesicle radius increases), we expect that the system to behave more like the 2D flat film case, in which the defects spontaneously emerge, move, and annihilate in a less ordered manner. This transition can be characterized by a dimensionless number $\gamma = \zeta D^2 / K$, where D is the vesicle diameter, K is the elastic constant and ζ is the activity. Further experimental and theoretical studies are needed to characterize this transition.

The manuscript presentation could be improved by considering following minor suggestions.

1. In Figure 3c3 the authors plot the defect energy. This quantity should be defined and described in more detail.

Response:

We thank the Referee for pointing out this problem. The defect (phase) energy is defined as the Landau-de Gennes short-range free energy, which governs the nematic-isotropic phase transition. The expression is provided by Eq. (2). We have added this description to the Results section in the revised manuscript, highlighted in blue.

2. It would be useful to discuss and compare the results of full numerical calculation to the much simpler model that treats motile defects as active repulsive particles. Is the simpler particle based model valid in any regime?

Response:

Following the Referee's suggestion, we have added and highlighted the following remarks to the revised manuscript regarding the comparison of our results to the simpler model: Our simulation results at low activity agree with the coarse-grained model proposed in Ref. 24 (cited as Ref. 25 in the revised manuscript) in the following respects: (1) Both models predict a ratchet-like shape when represented in a α -plot. As explained above, that shape is manifestation of the interplay between activity and elasticity. (2) Both models exhibit a threshold/onset activity, below which the system cannot overcome the elasticity to enter the oscillatory dynamic state. However, the defect trajectories predicted by the two models are different. In the particle-based model, the defects form pairs, and the paired defects revolve around the pair's center of mass. In contrast, in our simulations (and in agreement with experiments), the defect trajectories exhibit a more complicated behavior: the defects do not form pairs, and their trajectories are not simple circles. Instead, the defects can occasionally turn by approximately 90° during motion.

3. Is there a critical lower activity required for the onset of oscillations? If yes, what happens below this lower activity.

Response:

We thank the Referee for this question. There is an onset activity below which the defect configuration deforms and is balanced by the elasticity. We have emphasized this in the main text, highlighted in blue.

4. On line 181 authors discuss short arc and long arc states. Perhaps these can be defined somewhere.

Response:

Following the Referee's suggestion, we have added and emphasized the definitions of short-arc and long-arc state in the Results section, which are highlighted in blue.

5. One page 10 authors state: "We propose that the planar mode has high symmetry, and therefore the penalty arising from bending is at a local minimum." Perhaps this can be explained in more detail.

Response:

We thank the Referee for pointing out this problem. We have rephrased the statement to the following: "...in the excited state (the planar mode), the splay energy reaches its peak value, and the bend energy is relatively high".

6. Do authors examine the influence of elastic constants that differ in magnitude? If not this is another area worthy of future investigation.

Response:

We thank the Referee for this remark. It is also related to a question Reviewer #2 has asked: how biopolymer flexibility affects the dynamics. In the literature, a one-elastic-constant approximation has been widely adopted to simplify the theoretical analysis; some important physics, however, are missing from that representation. We have measured the splay and bend

elastic constants for certain active nematic systems. Our measurements indicate that for typical biopolymers, the bend constant is smaller than the splay constant. Following the Referee's suggestion, we have examined in detail the actual shapes of the defects that arise as the ratio of K_{33}/K_{11} is altered. As shown in the figure below, the director field surrounding the $+1/2$ defect is highly sensitive to the ratio of the elastic constants, specifically K_{33}/K_{11} . In fact, one can use experimental images of the defects to back out the numerical value of that ratio. Following the Referee's suggestion, we have done that for actin-myosin II systems of different filament lengths. The results are shown in Figure 1 below. These results, however, are beyond the scope of the current manuscript and have not been included in our work.

Figure 1. Left: analytical solution of the defect morphology, depending on the elastic-constant ratio of splay (K_{11}) and bend (K_{33}). Middle and right: experimental images of actin-Myosin II system with different filament lengths. The middle figure has longer filament length and our analysis indicates that $K_{11} \approx K_{33}$. The right figure has shorter filament length, and we find that $K_{33}/K_{11} \approx 0.3$.

Reviewer #2 (Remarks to the Author):

The authors use a continuum model of an active nematic to model active flow in a shell. This simulation approach has been used to describe active nematics in several contexts and gives good agreement with experiments. What is new here is handling the shell geometry within this simulation approach.

The work is motivated by experiments where microtubules and kinesin motors are confined on the surface of a vesicle. The combination of nematic order and the vesicle geometry imposes four topological defects on the surface of the vesicle and the activity drives these to move around the shell in orbits that are regular at low activities but become chaotic at high activities.

The current paper reproduces this behaviour well and gives many details of the competition between elasticity, flow and activity that control the defect dynamics. Defect trajectories are plotted as a function of activity and the balance between the various contributions to the free energy are explained. So this is a useful addition to the literature and a very careful piece of work. However, I think it would be much better published in a more specialised journal as the physics underlying the defect dynamics is explained in refs [6] and [24]. Moreover, there is an overlapping paper that the authors have missed: Motility of active fluid drops on surfaces, Diana

Khoromskaia and Gareth P. Alexander Phys. Rev. E 92, 062311 (2015).

Response:

We appreciate the Referee's comments and his/her perspective. We disagree, however, with the comment that the physics underlying the defect dynamics have been explained in refs [6] and [24] (cited as [25] in the revised manuscript). What was done in those references was to introduce a phenomenological framework to interpret some experimental observations. An explanation of the root causes for the observed behavior was not provided. In fact, the treatment offered in those two references cannot predict the behavior of the system from fundamental structural and material-property considerations. Furthermore, as we now emphasize in our revised manuscript, when that framework is used to examine the dynamics of the defects, it leads to incorrect results. Specifically, the model proposed in Ref. [24] (cited as Ref. [25] in the revised manuscript) predicts that the defects move in pairs, and that the paired defects revolve with respect to the center of mass of the pair. However, experiments show that the defects do not form pairs, and their trajectories are not simple circles. Instead, they can turn by approximately 90° during motion. Our model, which is based on a structural description of the liquid crystal, is able to capture that feature, as well as other experimental observations. We would also like to point out that our manuscript includes multiple predictions for new physics that future experiments will be able to address (e.g. the existence of a chaotic state or the emergence of a stagnation point for contractile systems).

We thank the Referee for the suggestion to cite the recent paper by D. Khoromskaia and G. P. Alexander. That paper is now cited in our revised manuscript. Our discussion of that work is brief because it refers to a different system, namely a drop on a flat surface that exhibits self-propulsion. It does not address in any way the dynamics of the defects observed in Keber et al.'s experiments, which is the subject of our manuscript. Finally, that reference relies on an approximate analytical solution to a free energy model in terms of a director (vectorial) representation, as opposed to the full theory for the tensorial order tensor that is more appropriate for a complete description of defects and their motion (including their full coupling to hydrodynamics).

Minor points:

1. Does the thickness of the shell map onto the thickness of the vesicle, and does it affect the results?

Response:

We thank the Referee for raising this question. Given the fact that the shell system is quasi-2D, the choice of our shell thickness should be very thin, to match the experiments and to ensure that the tetrahedral configuration is the globally stable state. By defining the shell thickness h and the vesicle diameter D , we can use the ratio h/D to characterize the relative thickness. Thus in our

simulations, we have studied both the $\frac{h}{D} = 0.1$ and $\frac{h}{D} = 0.05$ cases. Both systems show quasi-2D behavior and there is no qualitative difference between them.

2. Why is the material flow aligning, would flow tumbling dynamics affect the results?

Response:

We thank the Referee for this question. The orientation of nematic liquid crystals in a shear flow depends on the direction of preferential molecular alignment, the director, and the flow direction. If the director is aligned in the shear plane prior to flow, it would align with an angle θ to the flow direction. The angle is determined by $\theta = 0.5 \cos^{-1}(1/\lambda)$, where $\lambda = -\gamma_2/\gamma_1$, with γ_1 and γ_2 being the phenomenological volume torque coefficients, which have units of dynamic viscosity. When $|\lambda| < 1$, there is no steady-state solution for θ , thus the director rotates continuously in the shear plane. Such state is the so-called flow-tumbling regime. It is believed that for prolate nematogens (namely the constituents of the nematic phase), $|\lambda| > 1$. For the biopolymers considered in our manuscript, the aspect ratio (filament length/filament width) is about 60, deep in the prolate-nematogen regime. In this case, the material should always be flow-aligning. If the material happened to be in the flow-tumbling regime, the dynamics would be totally different from what is observed in experiments and in our manuscript. We have added a comment in the Results section related to this issue, highlighted in blue.

3. The authors refer to their model as a 'molecular' model. I would rather call it a coarse-grained or continuum model as there is no molecular detail.

Response:

Following the Referee's suggestion, we have changed the 'molecular model' to 'continuum model'. This change is highlighted in the main text.

4. It would be interesting if the authors could comment on the likely effect of the flexibility of the microtubules which is not included in the model.

Response:

We thank the Referee for this comment. We think that the filament flexibility would reduce the bend elastic constant. So the bend-instability of the extensile system should be more pronounced for flexible filaments. When activity is sufficiently high, the filaments may buckle, and our current model would not be able to describe the phenomena that ensue. We are currently interested in developing a better way of handling flexibility in our models, and are also working on a new manuscript discussing the possible effects of disparate elastic constants. We have added a small discussion regarding this comment in the Results section, highlighted in blue.

5. ... and on the flexibility of the vesicle - in the experiments protrusions sometimes form at the defect sites and it would be interesting (although difficult and not expected in the current M/S) to understand why.

Response:

We thank the Referee for raising this question. We think this phenomenon arises from the interplay between the dynamics of the +1/2 defect, the curvature of the vesicle, and the excess

surface area provided by the hypertonic stress. As the comet-like $+1/2$ defect moves, the microtubules at the tail of the defect move along the tangential plane of the vesicle, but the curvature forces the motion to bend, and follow the sphere's surface. This yields an outward stress that is able to protrude the vesicle.

6. p2 It is worth noting that nematic symmetry can arise from the flow field, not necessarily the shape of the active elements.

Response:

We thank the Referee for making this point. We have rephrased the sentence to avoid that misleading message. It is highlighted in the introduction.

7. p2 I do not agree with the statement 'confinement can induce and stabilise nematic flows' - confinement can stabilise the flow, but not induce it.

Response:

Following the Referee's comments, we have rephrased the statement in the following way: ``confinement can shape and stabilize the flow''. It is highlighted in the introduction.

8. ref [24] typo in Dogic

Response:

Following the Referee's suggestion, we have fixed the typo, and is highlighted in the bibliography.

Reviewer #3 (Remarks to the Author):

In the present manuscript the authors simulate an active nematic shell. With the help of a hybrid Boltzmann method they are able to simulate different regimes of activity. In the low activity state they recover amazingly precisely the activity observed in experiments, at higher activity they observe a chaotic regime. They are able to distinguish the different contributions to the free energy of the system - it is the interplay of enthalpy, bend and splay modes which gives rise to the unique dynamics.

The manuscript is beautifully written, and the approach as well as the results are bathing the way for new exciting studies of this class of materials. I do recommend enthusiastically the acceptance of the manuscript as it is. I do not see how to improve the manuscript any further.

Response:

We appreciate the Referee's positive and complimentary comments.

We thank again the reviewers and editor(s) for the time they invested in our manuscript.